

# Evaluation of extracellular polymeric substances extracted from waste activated sludge as a renewable corrosion inhibitor

Liew Chien Go[1], William Holmes[2], Dilip Depan[1] and
Rafael Hernandez[1,2]

[1] Department of Chemical Engineering, University of Louisiana at Lafayette, Lafayette, LA, USA
[2] Energy Institute of Louisiana, University of Louisiana at Lafayette, Lafayette, LA, USA

## ABSTRACT

**Background:** Waste activated sludge (WAS) has recently gained attention as a feedstock for resource recovery. The aim of this study is to investigate the corrosion inhibition efficiencies of extracellular polymeric substances (EPS) extracted from WAS.

**Methods:** The studied corrosion inhibitors were tested with carbon steel in 3.64% NaCl saturated with $CO_2$ at 25 °C, which is the typical oilfield environment. They were first prepared by EPS extraction (heating at 80 °C), followed by centrifugation for solid and liquid separation, then the supernatant was freeze-thawed five times for sterilization of microorganisms in WAS to terminate metabolic activities in the test inhibitors to ensure consistency in corrosion inhibition. The EPS mixture (supernatant) was then deemed as the test corrosion inhibitor. The inhibition performance was determined using potentiodynamic polarization scans.

**Results:** Waste activated sludge alone showed unsatisfactory inhibition. However, EPS extracted from WAS showed an optimum inhibition of approximately 80% with 1,000 mg/L of inhibitor. The average total solid (TS) and EPS contents of the WAS were 7,330 mg TS/L WAS and 110 mg EPS/g TS, respectively. Three sets of extracted EPS were scanned with fourier-transform infrared spectroscopy (FTIR) and showed almost overlapping curves, yielding the consistent inhibition performance.

**Discussion:** The potentiodynamic polarization results indicated that EPS acts as a mixed-type inhibitor which inhibits corrosion on both anode and cathode sites of metal surfaces. Based on the FTIR results, it was assumed that major chemical groups O–H, N–H, C–N, C=O, and C–H contributed to the inhibition by adsorbing on the metal surface, forming a biofilm that acts as a protective barrier to isolate the metal from its corrosive environment. Results show that WAS EPS corrosion inhibitors have inhibition performance comparable to commercial products, signifying their potential in commercialization. This corrosion inhibitor is renewable, biodegradable, non-toxic, and free from heavy metal, making it a superior green corrosion inhibitor candidate. Additionally, turning biomass into value-added product can be beneficial to the environment and, in this case, deriving new materials from WAS could also transform the economics of wastewater treatment operations.

Corresponding author
Rafael Hernandez,
rhernandez@louisiana.edu

## INTRODUCTION

The United States Environmental Protection Agency (EPA) is urging wastewater treatment facilities to be viewed as Renewable Resource Recovery Facilities that produce clean water, recover energy, and generate nutrients (*Capuco, 2013*). Efforts on energy recovery such as waste incineration and/or biogas production using anaerobic digestion are in practice at many wastewater treatment plants in the nation. However, further research is needed on the recovery of resources and nutrients using the biomass produced from wastewater treatment facilities. The Water Environment Federation believes that wastewater treatment plants are not just waste disposal facilities, but rather water resource recovery facilities that produce clean water and recover nutrients (*Water Environment Federation, 2019*). These facilities carry an array of high carbon sources as well as nutrients such as nitrogen and phosphorus-based compounds. By recovering these biomasses to value-added products, such facilities have a high potential to reduce the nation's dependence on fossil fuels through the production and use of renewable bioproducts. A great number of potentially marketable products could be recovered from the same facilities. Some value-added products derived from wastewater have been studied, like biodiesel, biodegradable plastics, adhesives, and enzymes useful in biomedical applications (*Capuco, 2013*).

Activated sludge is a common secondary treatment process for domestic and industrial wastewaters. This process is based on organics degradation by microorganisms under aerobic conditions. Carbons are used by the microorganisms to synthesize new cells or derive energy, leaving a relatively clear effluent. Some of the microorganisms are recycled to the aeration tank as a way of controlling the solids' residence time, while a fraction of the microorganisms is wasted and labeled as waste activated sludge (WAS). Millions of tons of this material are wasted annually by wastewater treatment operations. Extracellular polymeric substances (EPS) are the metabolic products produced by microorganisms that accumulate on their surface, forming slimy layers of biofilm. They are generally used to protect the cells against their external environment and to serve as carbon and energy reserves during starvation (*Lin et al., 2009*).

Studies on the generation of new materials like corrosion inhibitors by microorganisms are relatively scarce as compared to other green phytochemical-based corrosion inhibitors generated from agricultural residues. Pure cultures of sulfate-reducing bacteria, Pseudomonas (*Stadler et al., 2008*), Pseudomonas putida, and Pseudomonas mendocina KR1 (*Jayaraman et al., 1997*) have been tested and showed potential in corrosion inhibition. In the presence of some of these bacteria, with capability of developing biofilm to various degrees, it was observed that the mass loss of the tested metal can be decreased up to 15-fold. A similar study on sewage sludge extracted amino acids suggested the potential of sludge as a feedstock for corrosion inhibitor production (*Su et al., 2014*). Other recent studies have shown that EPS can adsorb heavy metals for the purpose of water treatment (*Liu, Lam & Fang, 2001*; *Wang et al., 2014*), showing that EPS possess distinct metal ion binding capability.

This study hypothesized that the metal binding characteristics of EPS can also act as a corrosion inhibitor when this EPS is extracted from WAS. Since WAS is a readily available

waste biological material source of wastewater treatment operations, extracting EPS directly from WAS can transform this waste biomass into a valuable product and reduce the environmental management challenge of WAS disposal.

As mentioned above, the hypothesis of this study is that a corrosion inhibitor can be made with EPS extracted from WAS. EPS can bind with the metal to form a biofilm layer on the metal surface to isolate the metal from its corrosive surroundings, thus promoting corrosion inhibition. To our knowledge, there are no previous studies in the peer reviewed literature on the evaluation of EPS extracted from WAS as a corrosion inhibitor. The objective of this study was to investigate the performance of WAS extracted EPS as an oilfield corrosion inhibitor. Oilfield conditions were simulated using carbon steel in 3.64% sodium chloride solution saturated with carbon dioxide gas using a potentiodynamic polarization technique. EPS was extracted by heating, using techniques described elsewhere (*Comte, Guibaud & Baudu, 2006*).

## MATERIALS AND METHODS

### Metal specimen preparation

Potentiodynamic polarization scans were performed on carbon steels of the following weight percentage composition: 0.17 C, 0.08 Mn, 0.014 P, 0.002 S, 0.022 Si, 0.02 Cu, 0.01 Ni, 0.04 Cr, 0.002 Sn, 0.042 Al, 0.006 N, 0.001 V, 0.0001 B, 0.001 Ti, 0.001 Cb, and the remainder iron. The pre-treatment of the specimens' surface was carried out by grinding with sand papers of 40, 220, 320 grits, rinsing with deionized water, and drying with paper towel. The specimens were used right away after the pre-treatment.

### Corrosive medium preparation

The test solution 3.64% of NaCl was used in the experiments. These solutions were prepared using deionized water and NaCl (Fisher Scientific, Hampton, NH, USA). Prior to initiating each experiment, $CO_2$ was sparged at 207 kPa (30 psi) in the test solution for 30 min. Then, the solution was transferred to the reactor and $CO_2$ sparging continued over the remainder of the experiment at about 138 kPa (20 psi).

### WAS samples processing

All WAS samples used for this study were collected from the recirculation stream of the aeration basins of East Wastewater Treatment Plant, Lafayette, LA 70501, USA. Three total samples were collected on three different days, that is, June 22nd, June 30th, and July 2nd of 2016, in order to reduce bias from variation of wastewater.

The WAS samples collected from each sampling day were divided into two groups. The first group of samples were used for corrosion inhibitor preparation, while the second group of samples were used for the quantification of WAS total solid (TS) and EPS.

### Corrosion inhibitors preparation

All WAS samples were concentrated by removing 80% of the water. The WAS samples were processed using two different methods in this study as control and test corrosion inhibitors. Hence, they were separated into two batches according to their processing

method. Control experiments were done without EPS extraction, while EPS was extracted and tested as corrosion inhibitors.

## Control experiments: WAS as corrosion inhibitors

A Welch Dry Fast Ultra Diaphragm Pump 2032 vacuum pump was used to evaporate 80% of the water from the WAS samples at room temperature, followed by centrifugation for 5 min at a relative centrifugal force of $1,207 \times g$ with a Thermo Scientific™ Sorvall™ ST 40 Centrifuge Series. The solid part was discarded, while all supernatants were collected and frozen for 24 h at $-20$ °C, then thawed for another 24 h. The freeze-thaw process was repeated five times to terminate the metabolic activities of the microorganisms. Then, the supernatant was used as a potential corrosion inhibitor.

## Corrosion inhibitor testing experiments: EPS as corrosion inhibitors

The WAS samples were heated at 80 °C and stirred at 600 rpm using a Fisher Scientific™ Isotemp™ Digital Stirring Hotplate. The samples were heated for EPS extraction and to evaporate 80% of water from the WAS samples. Heating is a physical method of EPS extraction. Furthermore, the relatively low temperature of heating minimizes the degradation of temperature labile compounds. Samples were then cooled down to room temperature and centrifuged at $1,207 \times g$ for 5 min. Then, the supernatant was collected and frozen for 24 h at $-20$ °C, followed by thawing for another 24 h. The freeze-thaw process was repeated five times before the supernatant was deemed as an EPS mixture for corrosion inhibition testing.

## Quantification of WAS total solid and EPS contents

Waste activated sludge samples were also used for quantification of TS content and EPS. Samples for TS content quantification were placed in ten 50 mL Corning centrifugal tubes. These tubes were centrifuged at $1,207 \times g$ for 5 min. Then, the supernatant was discarded. The solid left in the tubes was frozen at $-20$ °C overnight. The frozen solid was freeze-dried for 24 h using a Labconco freeze dryer at $-80$ °C and approximately 0 Pa. The dried solid was then weighed. The TS content was calculated based on the total weight of dried solid divided by 0.5 L of WAS.

For EPS quantification, a total of 500 mL WAS was divided into two 250 mL beakers. Both beakers were heated at 80 °C and stirred at 600 rpm for EPS extraction and to evaporate 80% of the water in the beakers. Separating WAS into two smaller containers and constant stirring were done to ensure even heating. These solutions were then combined, cooled down to room temperature, and centrifuged at $1,207 \times g$ for 5 min. The supernatant was collected in a 400 mL beaker and 100 mL of chilled ethanol ($-20$ °C) was added. The precipitate, EPS, was filtered out from the solution, dried, and then weighed. The quantity of EPS was calculated based on the weight of the dried EPS divided by 0.5 L of WAS, as well as the weight of dried EPS divided by dried solid in 0.5 L of WAS.

## Potentiodynamic polarization method

Potentiodynamic polarization experiments were carried out with a Gamry Flexcell Critical Pitting Cell Kit, connected to a Gamry Potentiostat Interface 1000. The reference, counter,

and working electrodes used were saturated calomel electrode, graphite rod, and the metal specimen, respectively. The setup was equipped with a heating jacket connected to a TDC4 Omega temperature controller to maintain the test solution at a desired temperature of 25 °C. The Glas-Col GT Series stirrer was connected to the setup externally and adjusted to 50 rpm to get the desired shear and to ensure even heating. The working solution volume was one L. The working area of the metal specimens had a circular form of five $cm^2$.

The potentiodynamic polarization scans were carried out in the potential range of −0.25 to +0.25 V vs. corrosion potential ($E_{corr}$) at a scan rate of 3 V/h. Corrosive medium was added into the reactor with $CO_2$ sparging constantly at 138 kPa (20 psi) throughout the experiment. The reactor was equilibrated for 30 min prior to the beginning of experiment. After the system was equilibrated, Tafel plots were graphed with Gamry DC105 DC Corrosion Technique Software until three relatively similar readings were obtained. Next, corrosion inhibitor was added into the reactor. The reactor was again allowed to equilibrate for 30 min, then Tafel plots were graphed. This step was repeated until three consecutive graphs with similar trends were observed to ensure the stability of the system.

The Tafel plot was plotted with the mean values of corrosion potential ($E_{corr}$) and corrosion current density ($I_{corr}$) from the triplicates of the experiments, while the electrochemical parameters obtained from the curves were reported with mean and standard deviation. The corrosion current densities were found by extrapolating the linear Tafel segment of the anodic and cathodic curves to the corrosion potential. The corrosion inhibition efficiency was then calculated with:

$$\text{Inhibition efficiecny}(\%) = \frac{I_{\text{corr, uninhibited}} - I_{\text{corr, inhibited}}}{I_{\text{corr, uninhibited}}} \times 100\% \quad (1)$$

## Fourier-transform infrared spectroscopy

Agilent Cary 630 fourier-transform infrared spectroscopy (FTIR) incorporated with MicroLab software were used for the FTIR analysis in this study. This equipment worked based on the Attenuated Total Reflection Method. The scanning was ranged between 4,000 and 400 $cm^{-1}$ with resolution of four $cm^{-1}$.

## RESULTS

### Control experiments: WAS as corrosion inhibitors

The Tafel plot obtained for this concentrated WAS is shown in Fig. 1. The electrochemical parameters obtained from the Tafel plot such as the values of corrosion potential, $E_{corr}$, corrosion current density, $I_{corr}$, and corrosion protection efficiency are presented in Table 1.

### Corrosion inhibitor testing experiments: EPS as corrosion inhibitors

Tafel curves generated from the potentiodynamic polarization measurements for carbon steel in 3.64% NaCl saturated with $CO_2$ in the absence and presence of some concentrations of test inhibitor at 25 °C are presented in Fig. 2. The details of

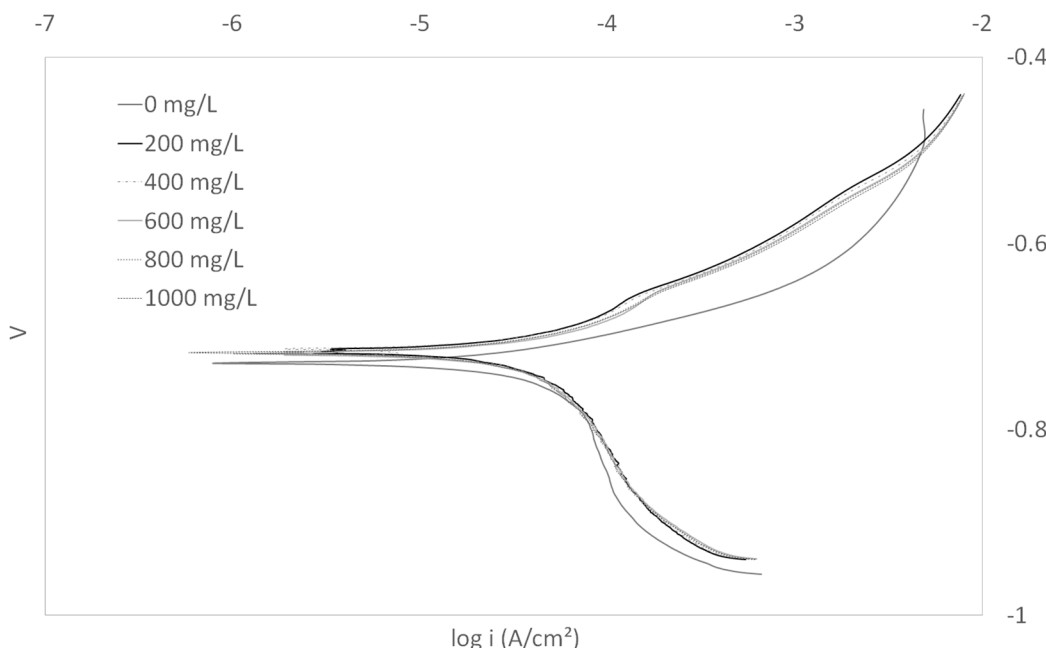

**Figure 1 Tafel plot for carbon steel in 3.64% NaCl concentrated with CO$_2$ with different concentrations of WAS at 25 °C.**

**Table 1 Electrochemical parameters and the corresponding inhibition efficiencies of carbon steel in 3.64% NaCl saturated with CO$_2$ containing different concentrations of WAS.**

| Concentration (mg/L) | $E_{corr}$ (V) Average ± standard deviation | $I_{corr}$ (μA/cm$^2$) Average ± standard deviation | Inhibition efficiency (%) Average ± standard deviation |
|---|---|---|---|
| 0 | −0.73 ± 0.00 | 44.53 ± 5.21 | N/A |
| 200 | −0.72 ± 0.01 | 43.44 ± 7.88 | 2.86 ± 6.90 |
| 400 | −0.72 ± 0.01 | 43.44 ± 7.88 | 2.86 ± 6.90 |
| 600 | −0.72 ± 0.01 | 43.44 ± 7.88 | 2.86 ± 6.90 |
| 800 | −0.72 ± 0.01 | 43.44 ± 7.88 | 2.86 ± 6.90 |
| 1,000 | −0.72 ± 0.01 | 43.44 ± 7.88 | 2.86 ± 6.90 |

electrochemical parameters obtained from the curves, namely corrosion potential ($E_{corr}$), corrosion current density ($I_{corr}$), and inhibition efficiency, are listed in Table 2.

## Total solid and EPS contents

Total solid and EPS contents in WAS were quantified, and the results are summarized in Table 3. TS quantification was conducted before EPS extraction. The standard deviation was relatively low which demonstrates the reproducibility of the experimental procedures and consistency of WAS composition.

## FTIR

The extracted white powder from WAS, assumed to be EPS at the time of the corrosion inhibition studies, was examined using FTIR, as shown in Fig. 3. Although the WAS

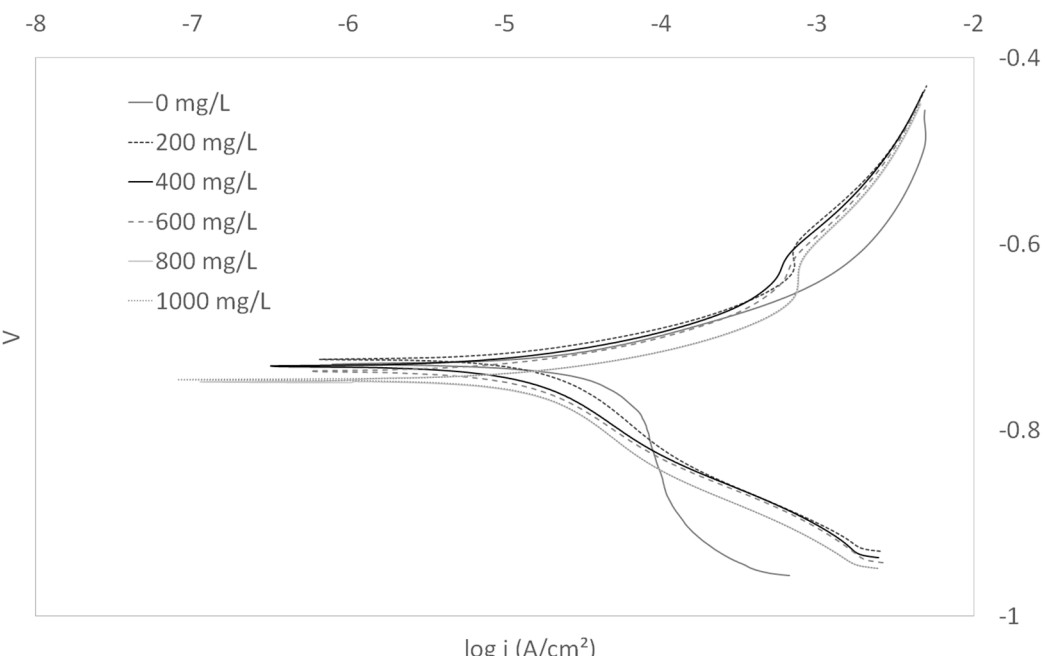

**Figure 2 Tafel plot for carbon steel in 3.64% NaCl concentrated with $CO_2$ with different concentrations of EPS mixture at 25 °C.**

**Table 2 Electrochemical parameters and the corresponding inhibition efficiencies carbon steel in 3.64% NaCl saturated with $CO_2$ containing different concentrations of EPS mixture.**

| Concentration (mg/L) | $E_{corr}$ (V) Average ± standard deviation | $I_{corr}$ (µA/cm²) Average ± standard deviation | Inhibition efficiency (%) Average ± standard deviation |
|---|---|---|---|
| 0 | −0.74 ± 0.00 | 46.49 ± 3.15 | N/A |
| 200 | −0.72 ± 0.00 | 15.62± 0.74 | 66.23 ± 3.71 |
| 400 | −0.73 ± 0.01 | 13.86 ± 1.48 | 70.00 ± 4.78 |
| 600 | −0.74 ± 0.00 | 13.56 ± 1.76 | 70.61 ± 5.34 |
| 800 | −0.75 ± 0.01 | 11.60± 1.97 | 74.78 ± 5.72 |
| 1,000 | −0.75 ± 0.01 | 9.75 ± 1.28 | 78.89 ± 3.72 |

**Table 3 Solid and EPS contents in WAS from three samples of different days.**

| Date | mg solid/L WAS | mg EPS/L WAS | mg EPS/g TS |
|---|---|---|---|
| 06/22/2016 | 7,500 | 1,340 | 110 |
| 06/30/2016 | 7,110 | 1,210 | 100 |
| 07/02/2016 | 7,370 | 1,270 | 110 |
| Average ± standard deviation | 7,330 ± 200 | 1,270 ± 70 | 110 ± 10 |

samples were collected from three different days, the FTIR curves of the EPS followed similar trends. This uniform EPS FTIR results explain the consistency of corrosion protection performance (low standard deviation values). The characteristic absorption of IR was tabulated in Table 4. These data were compared to a previous study on EPS extracted from WAS (*Comte, Guibaud & Baudu, 2006*). The functional groups of several

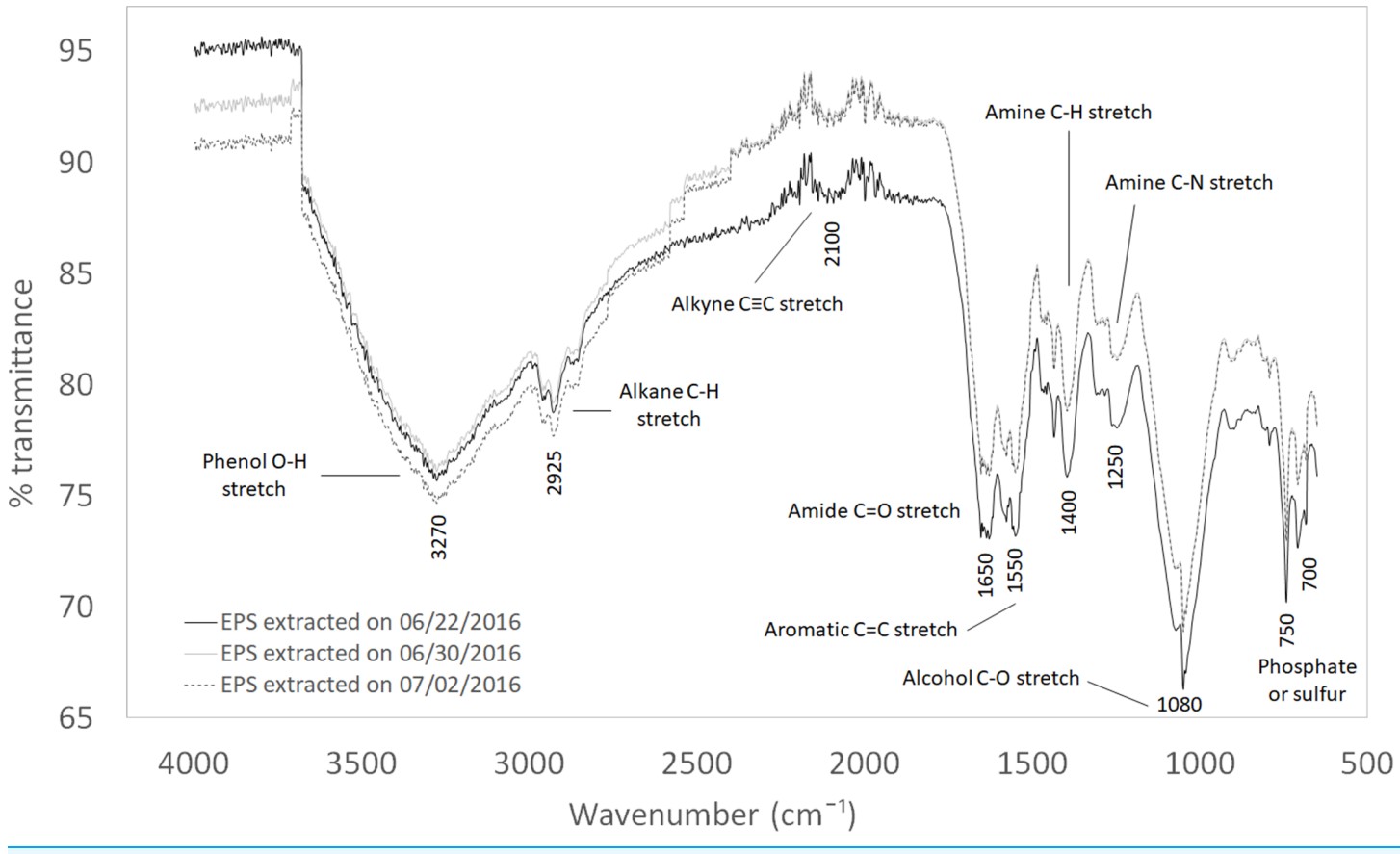

**Figure 3** FTIR of EPS extracted from different days.

**Table 4 Characteristic IR absorption frequencies of organic functional groups.**

| Characteristic absorptions (cm$^{-1}$) | Vibration type | Functional type |
|---|---|---|
| 3,200–3,600 | Phenol O–H stretch | OH into polymeric compounds |
| 2,850–3,000 | Alkane C–H stretch | |
| 1,690–1,630 | Amide C=O stretch | Proteins |
| 1,590–1,650 | Amide (I) N–H bend | |
| 1,500–1,560 | Amide (II) N–H bend | |
| 1,350–1,480 | Alkane C–H bending | |
| 1,080–1,360 | Amine C–N stretch | |
| <1,000 | Several visible bands | Phosphate or sulfur functional groups |

major compounds typically found in EPS were also identified, that is, O–H, N–H, C–N, C=O, and C–H groups.

# DISCUSSION

## Control experiments: WAS as corrosion inhibitors

The objective of utilizing WAS as a control was to examine a concentrated solution of compounds containing atoms of interest for corrosion inhibition. In the Tafel plot, it can

be observed that the curves are very close to each other, regardless of the increase in the concentration of WAS (potential corrosion inhibitor). Table 1 clearly demonstrates that $I_{corr}$ values are not impacted by the WAS concentration, and consequently inhibition efficiency. Increasing WAS concentrations from 0 to a 1,000 mg/L did not change $I_{corr}$. The data clearly indicate that the WAS tested did not contain compounds capable of inhibiting corrosion. Changes were so small that it was difficult to precisely measure corrosion inhibition. This difficulty resulted in relatively large and inconsistent standard deviations. The unsatisfactory results from both $I_{corr}$ and $E$ revealed that WAS was not suitable to be used as a corrosion inhibitor, rendering the value of $E_{corr}$ meaningless. Therefore, it can be deduced that the supernatant of WAS is not an effective corrosion inhibitor because of the absence or extremely diluted corrosion inhibitive compounds, making analysis for chemical oxygen demand unimportant. The supernatant was tested as a control to demonstrate the effectiveness and significance of the EPS extraction and performance as corrosion inhibitor.

## Corrosion inhibitor testing experiments: EPS as corrosion inhibitors

The curves revealed well defined anodic and cathodic polarization Tafel regions. From Fig. 2, it can be observed that both anodic and cathodic branch of polarization curves shift to lower values of current density as compared to the blank. This phenomenon indicated inhibition of both the hydrogen evolution and metal dissolution reactions (Morad, 1999). This may be ascribed to adsorption of inhibitor over the corroded surface (Motalebi et al., 2012). The corrosion inhibition mechanism of EPS is often described by: (1) the formation of a protective passive layer on metal the surface, (2) the depletion of oxygen by the metabolism of microorganisms, and (3) growth inhibition of corrosion-causing microbes, for example, from antimicrobial production by non-corrosive microorganisms; in multispecies biofilm, a combination of the different mechanisms may occur (Stadler et al., 2008; Jayaraman et al., 1997; Garrett, Bhakoo & Zhang, 2008; Kip & Van Veen, 2015). However, based on the results of $E_{corr}$, the corrosion inhibition mechanism of EPS in this study can be described by the formation of a protective passive layer (EPS biofilm) on the metal surface. There was no definite trend observed in $E_{corr}$ values in the presence of different concentrations of WAS in the experiments. This result indicated that the WAS may be regarded as a mixed type corrosion inhibitor (Abdallah, 2004) in presence of $CO_2$ saturated 3.64% NaCl solutions. Also, the maximum displacement in $E_{corr}$ of less than 0.085 V suggests the mixed mode of inhibition (Ma, Zhou & Sun, 2018). Mixed type corrosion inhibitor suppresses both anodic and cathodic corrosion reactions, typically indicating that the inhibitor adsorbed on the metal surface, forming a film to isolate the metal surface from the corrosive environment (Zhang, 2011). The application of EPS for the purpose of mitigating metal corrosion has been evaluated by other investigators (Stadler et al., 2008; Jayaraman et al., 1997). However, to our knowledge this is the first study evaluating EPS from a mixed microbial consortium (e.g., WAS), considered a waste and generated at millions of tons levels annually.

The maximal inhibition protection was 78.89% at a concentration of 1,000 mg/L. The inhibitor mixture was freeze-thawed at −20 °C five times to stop metabolic activity.

At a temperature of −2 °C, microbial growth can be stopped, and susceptible organisms can be killed (*Lumen Learning, 2019*). Elimination of microbial activity ensures a stable composition of the test solution. In the literature, research works have been conducted on utilizing the EPS of pure cultures of microorganisms as corrosion inhibitors for SAE 1018 in synthetic brine. Microorganisms such as Rhizobium meliloti 102F34, Streptomyces lividans TK23.1, and Bacillus circulans were grew with the metal coupons for a week. They showed corrosion inhibitions of 88%, 82%, and 73% corrosion inhibition, respectively (*Jayaraman et al., 1997*). A similar study utilizing WAS extracted amino acids showed 94% of inhibition when 372 mg/L of inhibitor was applied (*Su et al., 2014*). Amino acids are only a small fraction of the overall EPS composition. EPS is a mixture of extracellular proteins, polysaccharides, humic substances, uronic acid, and DNA. The amino acids in the extracellular proteins of EPS could have contributed to the corrosion inhibition. The presence of other compounds could have interfered with corrosion inhibition by adsorbing weakly on the metal surface or serving as a barrier (stearic hindrance) for the adsorption of chemical groups responsible for inhibition. Future studies will focus on the preparation of a mixture based specifically on the EPS groups that are known to perform as corrosion inhibitors.

For the case of commercial corrosion inhibitors, their corrosion protection performances are typically above 70%. Corrosion inhibitors developed from WAS have a corrosion inhibition performance that is within the range of commercial corrosion inhibitors. One advantage compared to pure culture investigations is that WAS is a readily available source in the wastewater treatment operations and does not require sterile conditions. The results obtained from this study strongly suggest the great potential of transforming WAS into a valuable material to inhibit corrosion issues in oilfield operations.

Furthermore, turning waste to value-added products can benefit the environment, society, and the economy. Instead of treating wastewater for waste management, value-added products such as corrosion inhibitors can be extracted from these readily available sources, transforming wastewater treatment operations into resource recovery units instead of sending WAS to the landfill or, even worse, incineration.

## Total solid and EPS contents

Total solid in WAS refers to all present solids. It is the residue remaining upon evaporation of WAS under 103 °C. The freeze-dry method was applied in this study to minimize the effect of heat during sample evaporation and prevent the loss of volatile solids.

Extracellular polymeric substances can be extracted from WAS by two means: chemical and physical methods. Some examples of chemical methods include the usage of EDTA, formaldehyde with NaOH, and glutaraldehyde; physical methods include heating, sonication, cation exchange resin, sonication with cation exchange resin, and centrifugation (*Comte, Guibaud & Baudu, 2006*; *Liu & Fang, 2002*). Typically, chemical methods give higher yield of EPS as compared to physical methods. However, a study by infrared analysis showed that chemicals used for extraction can contaminate

the mixture of EPS (*Comte, Guibaud & Baudu, 2006*). Therefore, chemical extraction was not considered in this study. Heating at 80 °C was chosen in this study because it is a well-established EPS extraction method in the literature. This temperature is considered mild for EPS extraction, preventing the degradation of EPS.

Even if the same extraction method is applied, the variation in sample preparation and modification in experimental procedure can also greatly affect the yield of EPS. For instance, by practicing heating extraction, some researchers extract EPS using raw WAS without any sample processing, while some researchers dry the WAS followed by resuspending the dried sludge solid in saline or water before EPS extraction. In addition, some researchers extract the EPS (heat) for 10 min, while some do it for hours.

Many studies report EPS content using different bases, such as mixed liquor suspended solid (MLSS) and volatile suspended solid (VSS). The range of EPS content can be tens to hundreds of mg EPS/g MLSS and mg EPS/g VSS. The EPS content found from this study was 110 mg EPS/g TS. *Comte, Guibaud & Baudu (2006)* extracted two batches of WAS EPS resulting in averages of 62 mg EPS/g VSS and 64 mg EPS/g VSS. *Liu & Fang (2002)* also extracted WAS EPS resulting in an average of 58 mg EPS/g VSS. Since the units are inconsistent, comparison of the EPS quantity can be difficult. Furthermore, EPS content in WAS is dependent on the wastewater facility operating conditions. However, knowing that VSS is typically about 78% of TS (*Ruiz-Hernando et al., 2015*), it can be estimated that the EPS content obtained from this study was about 86 mg EPS/g VSS. This estimate is about 25% compared to the literature. This could have been caused by the longer EPS extraction time. In general, the extraction time of other studies is approximately 30 min. We extracted EPS for 6 h, until 80% of the liquid from WAS was evaporated.

The compositions of WAS and EPS could vary due to wastewater treatment operating conditions and weather. Considering that WAS samples were only collected during summer, the TS and EPS contents were notably consistent. Since this study is a proof of concept to investigate if WAS EPS can act as a potential corrosion inhibitor, several other factors that can affect the WAS and EPS were not considered. For example, WAS seasonal variations or verification of microbial metabolic activity. Future work will focus on formulating a synthetic EPS mixture with known chemical compounds. The composition of natural EPS will be used as a basis. The hypothesis is that the new corrosion inhibitor formulation can act similarly or better to the studied natural EPS. By doing so, the variability in wastewater can be eliminated in addition to having a better understanding of the inhibition mechanism and inhibitor-metal interaction.

## FTIR

The composition of EPS is well established in the literature. EPS are high molecular weight mixtures of biopolymers which are mainly made up of extracellular proteins, polysaccharides (carbohydrates), and humic substances, with a small amount of uronic acid and DNA. Typically, carbohydrates have been identified as the major constituents in the EPS of many pure cultures (*Cescutti et al., 1999*; *Kennedy & Sutherland, 1996*), whereas

other researchers found relatively high protein concentrations in the sludge of many wastewater treatment reactors (*Liu & Fang, 2002*).

The composition of these assorted chemicals in EPS is heavily dependent on the extraction method. Typically, the EPS extracted by heating has the highest protein concentration, followed by carbohydrates, humic acid, nucleic acid, and uronic acid (*Comte, Guibaud & Baudu, 2006*; *Liu & Fang, 2002*; *Fang & Jia, 1996*).

The FTIR results showed that functional groups O–H, N–H, C–N, C=O, and C–H were in the extracted EPS, suggesting the presence of protein, carbohydrates, humic substances, uronic acid, and DNA. Heating yields higher protein concentration, which contributes to a higher overall N–H functional group. Compared to a carbohydrate-rich mixture, a protein-rich mixture has a clear advantage due to a less drastic ratio of N–H to other functional groups. Having more variety of functional groups in a mixture could improve the overall corrosion inhibition due to synergistic effect (*Zhang et al., 2008*; *Gece & Bilgiç, 2010*; *Khaled, 2010*; *Eddy, 2011*). The corrosion inhibition mechanism of EPS can be inferred based on the present functional groups. They are rich in nitrogen and oxygen atoms, resembling typical organic corrosion inhibitors. Knowing that organic corrosion inhibitors are typically made up of a polar head that usually consists of nitrogen, oxygen, or sulfur atoms attached to a non-polar hydrocarbon chain, it can be deduced that the observed functional groups had contributed to the overall corrosion inhibition of carbon steel in 25 °C $CO_2$ saturated 3.64% NaCl solution.

Based on the basic corrosion theory, corrosion is a redox reaction. Slowing down either the reduction or oxidation reaction in a corrosion system can slow down the overall corrosion rate. The corrosion inhibition mechanism of EPS can be explained by the adsorption of various chemical compounds (corrosion inhibitors) on the metal surface. The functional groups rich in nitrogen and oxygen atoms acted as the polar head of organic corrosion inhibitors, adsorbing on the metal surface, while the non-polar hydrocarbon chain attached to the polar head isolated the metal surface from its corrosive surrounding, forming a protective biofilm. The biofilm reduced the oxidation of metal, thus reducing the overall corrosion rates.

## CONCLUSIONS

In this study, WAS and EPS extracted from WAS were tested as corrosion inhibitors. Non-heat-treated WAS did not inhibit corrosion at any of the evaluated concentrations. The heat treated (EPS extracted) WAS showed a corrosion inhibition performance of approximately 80.0%, which compares favorably with commercial corrosion inhibitors. Developing a corrosion inhibitor using a readily available source such as WAS is an example of waste conversion to a value-added product. It could add value to WAS and in combination with the generation of other products from WAS to transform the economy efficiency of wastewater treatment operations. Based on the results presented and the needs and requirements of corrosion protection service providers, the future direction of the current research is the following: (1) produce a corrosion inhibitor with consistent properties that can lead to consistent inhibition performance (2) reformulate a corrosion inhibitor mixture based on EPS composition that requires a lower inhibitor concentration.

This could be done by blending one compound from each major chemical group present in natural EPS, namely protein, carbohydrate, humic acid, nucleic acid, and uronic acid. Compound selection can be done by prioritizing those that have previously been studied as effective corrosion inhibitors or those containing chemical groups favoring inhibition performance, such as compounds with nitrogen, oxygen, or sulfur atoms, bigger molecular structures, and longer hydrocarbon tails.

### Funding

This work was supported by the Coastal Chemical Research Fund (Project No.: 14-1227). The funders had no role in study design, data collection and analysis, decision to publish, or preparation of the manuscript.

### Grant Disclosures

The following grant information was disclosed by the authors:
Coastal Chemical Research Fund: 14-1227.

### Competing Interests

The authors declare that they have no competing interests.

### Author Contributions

- Liew Chien Go conceived and designed the experiments, performed the experiments, analyzed the data, contributed reagents/materials/analysis tools, prepared figures and/or tables, authored or reviewed drafts of the paper, approved the final draft.
- William Holmes conceived and designed the experiments, contributed reagents/materials/analysis tools, authored or reviewed drafts of the paper, approved the final draft.
- Dilip Depan authored or reviewed drafts of the paper, approved the final draft.
- Rafael Hernandez conceived and designed the experiments, analyzed the data, contributed reagents/materials/analysis tools, authored or reviewed drafts of the paper, approved the final draft.

### Data Availability

The raw measurements for potentiodynamic polarization and FTIR are provided in a Supplemental File.

### Supplemental Information

Supplemental information for this article can be found online at http://dx.doi.org/10.7717/peerj.7193#supplemental-information.

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
