# Peer review of "Evaluation of extracellular polymeric substances extracted from waste activated sludge as a renewable corrosion inhibitor"

_PeerJ, doi:10.7717/peerj.7193_

## Round 0.1 · original submission · Major Revisions

Your submission has been reviewed by the experts in the field. Based on their comments, your paper is recommended for major revisions. Please make sure to include a point to point response in terms of your corrections or rebuttal towards the reviewers' comments.

Reviewer 1 ·

Basic reporting

The authors report a study on the use of EPS extracted from WAS as a corrosion inhibitor. The background was well articulated and research gaps are clearly identified. The manuscript is well written and organized. The hypothesis is well defined and experimental design and analysis were used to test the hypothesis. The results are new and interesting and support the hypothesis.

Experimental design

The research is novel and original. The experimental design was well used to plan the experiments. and the results were analyzed to support the hypothesis. The methodology is appropriate and comprehensive. The research question is interesting.

Validity of the findings

The hypothesis is well supported by the experimental results. Data are robust. Conclusions are valid.

Additional comments

Overall, this study is well planned and executed. The results are interesting and advance our knowledge on the new application of EPS from sludge for corrosion inhibition. Specific comments are listed below:

1. The centrifuge rpm should be converted into centrifugation force ( x g), as different types of centrifuges (different diameters) at the same rpm will have different centrifugation force (x g). The changes should be made in a number of places, such as lines 131 and 138 etc.
2. Line 266, "maximum" should be "maximal".
3. The quantity of EPS extracted from WAS seems very low, as compared to a number of EPS studies. The typical EPS amount is from tens of mg EPS/g MLSS to hundred mg EPS/gMLSS. Please provide an explanation of the low EPS extraction efficiency.
4. Fig 3, FTIR spectra. Please label the typical FTIR peaks with wave numbers.

Reviewer 2 ·

Basic reporting

- English is generally good
- Insufficient background material was provided. I have added that the authors go back and conduct a more thorough review of the relevant literature
- graphs need improvement, tables need to be revised with respect to significant figures
- results are relevant to hypothesis

Experimental design

- relevant to journal
- the methods are described in detail.

Validity of the findings

- The study is interesting. However, the authors seem unaware of the existing literature in their area.
- The logic for conducting some of the experiments will need to be explained better.
- The authors will need to do a better job in comparing their results with the existing literature and putting their work in context

Additional comments

Line 64: I think at some point in the document it should be mentioned that many wastewater treatment facilities have the means to recover energy from WAS through processes such as anaerobic digestion (biogas) and incineration (heat recovery).
Line 88: The authors claim that there is no literature on the use of EPS as a corrosion inhibitor. Here are some examples that have not been cited in this paper:
Su,W.; Tang, B.; Fu, F.; Huang, S.; Zhao, S.; Bin, L. A new insight into resource recovery of excess sewage
sludge: Feasibility of extracting mixed amino acids as an environment-friendly corrosion inhibitor for
industrial pickling. J. Hazard. Mater. 2014, 279, 38–45.
Line 93: slight modification… vague wording
Line 114: all samples were collected during summer season, what variabilities in the EPS composition are expected in the winter months?
Line 119: the wording of WAS solid is confusing, are we speaking here about the WAS TSS and VSS content? Line 150-157: this seems like a tedious method to measure TSS, also why are you not calling it TSS?
Line 133: how did you measure metabolic activity to make sure it was terminated?
Line 141: solvent extraction is a process and not a substance. Fix the writing here.
Line 158: So there are many papers on the use of amino acids on corrosion inhibition. The extraction method includes heating to 80 degrees. Do the authors expect that this would alter the amino acids?
Zhang, D.Q.; Cai, Q.R.; He, X.M.; Gao, L.X.; Zhou, G.D. Inhibition effect of some amino acids on copper
corrosion in HCl solution. Mater. Chem. Phys. 2008, 112, 353–358.
24. Gece, G.; Bilgiç, S. A theoretical study on the inhibition efficiencies of some amino acids as corrosion
inhibitors of nickel. Corros. Sci. 2010, 52, 3435–3443.
25. Khaled, K. Corrosion control of copper in nitric acid solutions using some amino acids—A combined
experimental and theoretical study. Corros. Sci. 2010, 52, 3225–3234.
26. Eddy, N.O. Experimental and theoretical studies on some amino acids and their potential activity as inhibitors
for the corrosion of mild steel, part 2. J. Adv. Res. 2011, 2, 35–47.
General comment: quality of graphs produced needs improvement before they can be published. Also, the curves can not be distinguished in black and white print. I suggest the authors use a combination of shades and dashed curves so that the curves can be reviewed properly.
General comment: I think the authors will need to do a more in-depth review of the relevant literature to strengthen their discussion and include a comprehensive review of the literature as it pertains to their subject and findings.
293-296: proper analysis of the FTIR results was not provided. This is a key component of the paper as it discusses the functional groups contributing to the corrosion inhibiting properties of EPS.
General comment: there has been no discussion on the oxidation-reduction activities in the biofilm and how this can impact the corrosion inhibition as a result of biofilm formation.
Line 226: what is the mechanism of the functional groups contributing to corrosion inhibition, this needs to be discussed.
Line 235: the “WAS samples” basically include supernatant from centrifuged WAS that was freeze-thawed a few times. Why were the authors thinking that this would have any potential corrosion inhibition in the first place? The logic is not clear to me. Centrifuging WAS samples and taking the supernatant will give you water with minimal carbon. Were there any COD tests conducted to determine the differences in the carbon content of supernatant collected from WAS samples and EPS samples? This needs to be included as the main corrosion inhibitor is the amino acids (which this discussion is also missing from the paper).
266-273: this part is missing comparisons with other studies and literature values. I think the authors should think a bit more about what the EPS composition is and do a literature search on the corrosion inhibiting properties of EPS constituents, then they will be able to find a breadth of literature for comparing their data.
275: WAS is not grown…
General comment: the paper can use some discussion on the various methods of EPS extraction and how that impacts the EPS composition and functional groups and also what the expected impacts are on the corrosion inhibition of the extracted EPS.
General comment: significant figures in tables vs. variability in samples and measurement method. 7.11 vs. just reporting as 7. This needs to be thought through and fixed in all tables

---

## Round 0.2 · accepted · Accept

I believe the revised manuscript has addressed the concerns raised by the reviewers. It will be sent for production.